# Single cell transcriptional analysis reveals novel innate immune cell types

Linda E. Kippner[1,3], Jinhee Kim[2,3], Greg Gibson[2] and Melissa L. Kemp[1]

[1] The Wallace H. Coulter Department of Biomedical Engineering, Georgia Institute of Technology and Emory University, Atlanta, GA, USA
[2] School of Biology, Georgia Institute of Technology, Atlanta, GA, USA
[3] These authors contributed equally to this work.

## ABSTRACT

Single-cell analysis has the potential to provide us with a host of new knowledge about biological systems, but it comes with the challenge of correctly interpreting the biological information. While emerging techniques have made it possible to measure inter-cellular variability at the transcriptome level, no consensus yet exists on the most appropriate method of data analysis of such single cell data. Methods for analysis of transcriptional data at the population level are well established but are not well suited to single cell analysis due to their dependence on population averages. In order to address this question, we have systematically tested combinations of methods for primary data analysis on single cell transcription data generated from two types of primary immune cells, neutrophils and T lymphocytes. Cells were obtained from healthy individuals, and single cell transcript expression data was obtained by a combination of single cell sorting and nanoscale quantitative real time PCR (qRT-PCR) for markers of cell type, intracellular signaling, and immune functionality. Gene expression analysis was focused on hierarchical clustering to determine the existence of cellular subgroups within the populations. Nine combinations of criteria for data exclusion and normalization were tested and evaluated. Bimodality in gene expression indicated the presence of cellular subgroups which were also revealed by data clustering. We observed evidence for two clearly defined cellular subtypes in the neutrophil populations and at least two in the T lymphocyte populations. When normalizing the data by different methods, we observed varying outcomes with corresponding interpretations of the biological characteristics of the cell populations. Normalization of the data by linear standardization taking into account technical effects such as plate effects, resulted in interpretations that most closely matched biological expectations. Single cell transcription profiling provides evidence of cellular subclasses in neutrophils and leukocytes that may be independent of traditional classifications based on cell surface markers. The choice of primary data analysis method had a substantial effect on the interpretation of the data. Adjustment for technical effects is critical to prevent misinterpretation of single cell transcript data.

Corresponding author
Melissa L. Kemp,
melissa.kemp@bme.gatech.edu

## INTRODUCTION

A growing body of evidence indicates that cell populations, even those comprised of genetically identical cells, can be highly phenotypically heterogeneous (*Enver et al., 2009*; *Niepel, Spencer & Sorger, 2009*; *Spencer et al., 2009*; *Spencer & Sorger, 2011*), and that these differences between individual cells can have functional consequences (*Feinerman et al., 2010*). Such non-genetic heterogeneity has been indicated in immune cell functionality (*Feinerman et al., 2010*; *Shalek et al., 2013*) and has also been suggested as a driving force of stem cell development and cell fate decisions, such as lineage choice in hematopoietic stem cells (*Chambers et al., 2007*; *Chang et al., 2008*; *Dietrich & Hiiragi, 2007*; *Kalmar et al., 2009*; *Kobayashi et al., 2009*; *Singh et al., 2007*; *Stockholm et al., 2007*). Cellular heterogeneity is also an underlying source of the development of phenotypically different subpopulations due to individual cell responses to changes in microenvironment within genetically identical populations (*Neildez-Nguyen et al., 2008*). Such functional subgroups can also have substantial pharmacological consequences, notably with regards to cancer treatment, where partial drug resistance in tumor cell populations poses a significant problem (*Cohen et al., 2008*; *Gascoigne & Taylor, 2008*; *Niepel, Spencer & Sorger, 2009*; *Orth et al., 2008*; *Sharma et al., 2010*; *Shi, Orth & Mitchison, 2008*). For example, non-genetic variations in response to pro-apoptotic stimuli have been found across several cell lines and stimuli, resulting in phenotypically different subgroups even within clonal cell populations (*Cohen et al., 2008*; *Gascoigne & Taylor, 2008*; *Geva-Zatorsky et al., 2006*; *Huang, Mitchison & Shi, 2010*; *Orth et al., 2008*; *Sharma et al., 2010*; *Shi, Orth & Mitchison, 2008*; *Spencer et al., 2009*). In light of this evidence, it is apparent that single cell resolution is needed in order to achieve systems level understanding of functionality.

It is becoming evident that established methods, whereby averaging population data essentially assumes that all cells within a population are equivalent, are vastly oversimplifying cell functionality and obscuring the presence of cellular subtypes (*Sachs et al., 2005*); however, a more detailed analysis has been hindered by technical limitations. Previously, transcription analysis has been constrained to population averages, due to the inability to quantify single cell levels of mRNA with existing techniques, such as such as northern blotting or classical qRT-PCR (*Flatz et al., 2011*; *Kalisky & Quake, 2011*; *Kurimoto et al., 2007*; *Shi et al., 2011*; *White et al., 2011*). Major technical advances in single cell measurement systems have now enabled the investigation of such cell-level information (*Huang et al., 2014*; *Janes et al., 2010*; *Morris, Singh & Eberwine, 2011*; *Rajan et al., 2011*; *Wang & Janes, 2013*; *Zhang et al., 2011*). These advances include high-throughput nanoscale real time PCR, which allows for mapping of transcriptional profiles by highly parallelized assays enabled by microfluidics.

Standard methods for processing qRT-PCR data are well established; however these methods are based on population averaged data and it cannot be taken for granted that the same approaches are optimal for single cell data. Indeed, single cell gene transcripts have been shown to follow log normal distribution curves (*Bengtsson et al., 2005*); thus, mean population averages are heavily influenced by a few cells showing relatively high expression levels. As single cell data is inherently noisy, this must be taken into account when choosing

**Peer**J ___________

analysis methods. For example, housekeeping genes show considerable variability of expression at the single cell level such that standard methods of data normalization based on such genes should not be used (*McDavid et al., 2013*). In addition, single cell measurements exhibit noise due to technical variability and this must ideally be accounted for without losing variability due to biological functionality, which is often at comparable levels (*Brennecke et al., 2013*). A particularly important consideration is whether the complete absence of signal is due to lack of expression or to stochastic technical failure. All analytical approaches make assumptions regarding this issue that could have a major impact on the conclusions derived from different modes of analysis (*McDavid et al., 2013*).

The biological motivation for the current study was to assess gene expression variability among single leukocytes, and whether the prevalence of functional sub-types (as defined by gene expression) varies among individuals. Neutrophils and T lymphocytes were selected as representatives of the innate and adaptive branches of the immune system, respectively. Recent studies have revealed a close correlation of functional phenotype to transcriptional profile (*Dalerba et al., 2011*; *Hoshida et al., 2008*; *Mucida et al., 2013*), and we hypothesized that our results would yield immune cell subclasses separated not only by the traditional surface markers, but also by intracellular signaling components, as well as other functional markers. As bimodality in expression of individual transcripts can be an indicator of functional heterogeneity (*Shalek et al., 2013*), we further asked whether cellular subclasses were defined by shared bimodality of multiple transcripts between cells. To that end, we performed gene expression pattern analysis and hierarchical clustering of our cell populations. We found that genes exhibiting bimodal distribution patterns were preferentially assigned to the same cell clusters in our data sets.

In overcoming the technical challenges of analyzing single cell data, we found that the decisions made in data processing can have dramatic consequences for the interpretation of cellular subpopulations. We systematically explore and recommend approaches that can be used in order to consistently analyze multiple single cells from multiple donor individuals across multiple genes. Nine alternate methods of data exclusion and normalization are considered, and their effect on secondary data analyses, such as hierarchical clustering, is assessed. Our results show that analysis and correct interpretation of single cell gene expression data is dependent on the method chosen for primary data analysis, specifically on the method chosen for data normalization.

## MATERIALS & METHODS

A schematic diagram of the workflow for the experiment as well as the data analyses described below is depicted in Fig. 1.

### Primary cell extraction and single-cell sorting

Neutrophils and T lymphocytes were extracted from 5 ml whole blood from 6 healthy donors and isolated based on phenotype by negative selection using magnetic beads (EasySep neutrophil extraction kit, Stem Cell Technologies, or Dynabeads untouched T cells, Life Technologies). One donor's neutrophil count was too low for further processing, therefore all results presented for neutrophils consist of $n = 5$. Negative

## A Data Acquisition Workflow

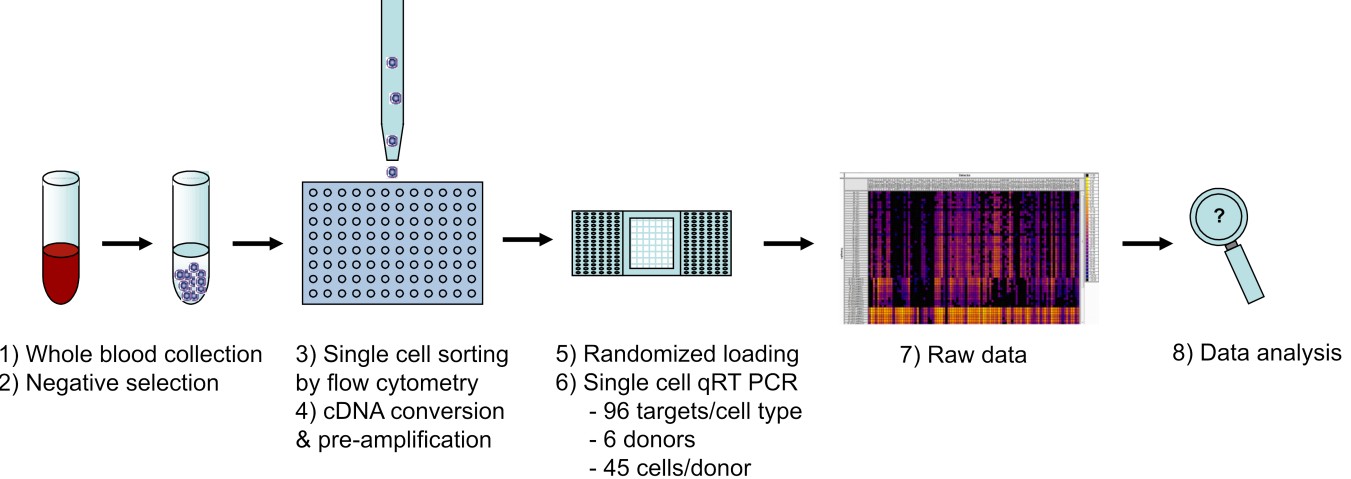

1) Whole blood collection
2) Negative selection

3) Single cell sorting
by flow cytometry
4) cDNA conversion
& pre-amplification

5) Randomized loading
6) Single cell qRT PCR
   - 96 targets/cell type
   - 6 donors
   - 45 cells/donor

7) Raw data

8) Data analysis

## B Data Processing Workflow

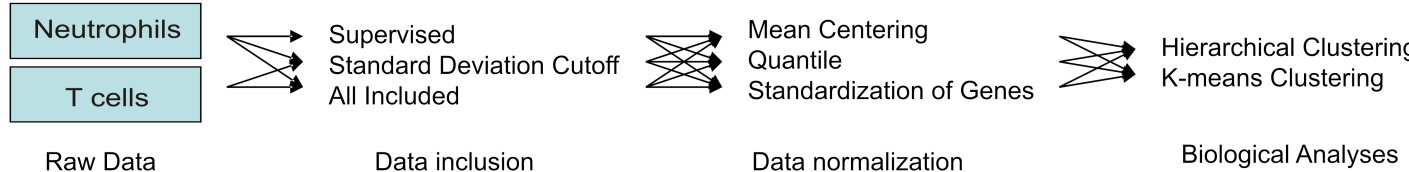

| Neutrophils | | Supervised | | Mean Centering | | Hierarchical Clustering |
| T cells | | Standard Deviation Cutoff | | Quantile | | K-means Clustering |
| | | All Included | | Standardization of Genes | | |

Raw Data      Data inclusion      Data normalization      Biological Analyses

**Figure 1 Workflow of single cell transcriptional data acquisition.** (A) Whole blood was collected from healthy donors, and negative selection used to isolate T cell and neutrophil populations. Single cell sorting was then used to deposit one cell per well into 96-well plates, pre-loaded with lysis buffer. Following this, cDNA conversion and pre-amplification was done in plate, and resulting cDNA samples randomly loaded onto microfluidic arrays. qRT-PCR reactions were run simultaneously against 96 gene targets per cell. Raw data was obtained as Ct values. (B) For data processing, three methods were tested for data inclusion in combination with three methods for data normalization. Following this, the resulting nine data sets were analyzed for biological information by gene expression pattern analysis, detection of cellular subtypes by hierarchical clustering, and comparison of individual donor subtype representation.

selection was chosen so as to avoid cellular activation due to receptor cross-linking. For each purified cell type, flow cytometry sorting with a BD FACS Aria II gated by forward- and side scatter was utilized to deposit single cells into a 96-well PCR plate preloaded with 5 µl of lysis buffer with 0.05U Superase RNase inhibitor (Life Technologies) per well. The plates were centrifuged for 1 min at 4 °C and immediately frozen and stored at −80 °C. All donors were individuals enrolled in The Center for Health Discovery and Well-Being at Emory Midtown Hospital and provided written consent for participation in the study. The protocol for blood collection was approved by the Georgia Tech Institutional Review Board (approval #H09364).

## Single cell qRT-PCR

The cellular lysates were converted to cDNA and 96 target genes per cell type were pre-amplified with a pool of 96 primer pairs targeting genes representing pattern recognition, cell-type markers, intracellular signaling, transcription, and immune response. For each donor, amplified cDNA samples from 48 cells of each type were then randomized and re-plated across 5 Fluidigm 96 × 96 microfluidic arrays, in order to avoid any plate effects confounding the analysis of single donors. Gene-specific quantitative real-time PCR reactions were performed using the Fluidigm BioMark I nano-scale platform. Negative controls (without cDNA) and samples of 10 and 100 cells were used as controls for single-cell loading. The mean difference in Ct value between 1 and 10 cells and between 10 and 100 cells per sample was determined in independent assays, providing a measurable control for single cell loading of each sample.

To enable reproducible comparison of gene expression between qRT-PCR samples, data is usually normalized with respect to data obtained for an internal or endogenous reference gene. Housekeeping genes such as $\beta$-actin and glyceraldehyde 3-phosphate dehydrogenase (GAPDH) are most often used because their expression levels are expected to remain constant. Unfortunately, single cells exhibit large heterogeneity in housekeeping gene expression levels, and this method cannot be used as control for reproducible comparison of gene expression between single cell samples (*Liss et al., 2001*; *Suzuki, Higgins & Crawford, 2000*).

## Quality control and data exclusion

In order to control for single-cell sensitivity and consistency in sample loading, single cell readings were compared to multi-cell controls. Mean expression levels from 10 randomly chosen single cell samples were calculated for each gene. The values obtained were compared by regression to the mean expression levels for the corresponding genes from the 10-cell samples, and $R^2$ values above 0.65 were observed in all cases, indicating good concordance between single cell and multicell measurements.

Raw data for gene expression were obtained as Ct values between 1 and 40, with lower Ct value indicating higher abundance of gene-specific product. Missing data points were coded as Ct values of 999; such values can either be due to null or very low expression of the target gene in question or due to a failed reaction (truly missing data). Single missing measures may indicate technical failures, but consistent absence of a similar set of lowly expressed transcripts is more likely to imply coordinated loss of expression. Downstream methods differ largely with respect to how the missing data is handled. Three different sets of criteria were used for data exclusion for each of the two (neutrophil and T lymphocyte) data sets.

(A) *Supervised Data Exclusion.* For the neutrophil data set, an empirical cutoff was set to transcripts present in at least 70% of cells, and subsequently to cell samples expressing at least 70% of these most uniformly expressed genes. We reasoned that the absence of the same set of genes in a common set of cells would imply true absence of expression, and used hierarchical clustering to provide a preliminary indication of such clusters  of

non-expressed genes. 23 such co-regulated low-abundance genes were identified, for which missing values were re-assigned a Ct value of 40 (the maximum number of cycles). Subsequently, for 36 genes, sporadic missing data was assumed to represent technical error and these values were reassigned to the average Ct for the gene in question in the data set. 34 genes and 18 cells were excluded in their entirety. Expression was evaluated for 59 genes in 202 cells. Because the T lymphocyte data set did not contain a natural cutoff for transcript presence, this method of analysis was not implemented for the T lymphocyte data.

(B) *Data exclusion based on median standard deviation cutoff*. All missing data values were initially set to Ct 40, and the mean Ct and number of missing data points were calculated for all genes. The second and third highest expressed genes in the data set were selected and their mean Ct and standard deviation calculated. (The highest expressed gene in both the neutrophil and T lymphocyte data set were treated as outliers and ignored for the purposes of calculating mean Ct, due to expression levels far higher than all other genes). Any gene whose average expression was within a cutoff of three standard deviations of the mean Ct value for the two chosen genes was included. All cells expressing less than half of these genes were then excluded. A plot of the maximum Ct across all cells for all 96 genes in the neutrophil data set showed a bimodal distribution of maximum Ct values, with a second peak starting at Ct 37 that corresponds to cells deemed not to express the target gene. The limit of detection (LOD) was thus set to Ct 37 for neutrophils, and the LOD Ct was set to 38 for T lymphocytes by the same methodology. All data values above LOD Ct, including Ct 999, were replaced with 37, and the LOD Ct value was then subtracted from all other Ct values, according to the $Log_2EX$ method ($Log_2EX = LOD$ Ct–Ct (*Abdel-Rahman et al., 2008*)). Consequently, the adjusted expression measure for this method is inverted and ranges from 0 to LOD Ct, with more highly expressed transcripts having higher values, more in line with intuition and with microarray or RNA-Seq data analysis. For the T cell data set, Ct values above LOD were interpreted and analyzed in two alternate ways; either as representing no expression of the target gene, with Ct values set to 0 or, alternatively, as missing data points due to technical error, with missing values replaced with average Ct for the gene (analogous to the Supervised data analysis method for neutrophils). Subsequently, entire cells were excluded, if the two gene targets with highest expression in our data set were more than three standard deviation units lower than the median. Additionally, any genes that were not expressed in any cell sample were excluded from the data set. For the T lymphocyte data set, 2 genes that were only expressed in one cell were also excluded from analysis. This resulted in the exclusion of 12 neutrophils and 31 genes in the neutrophil data set, and 7 T lymphocytes and 63 genes in the T lymphocyte data set. Expression was evaluated for 62 genes in 208 cells in the neutrophil data set and just 29 genes in 244 cells in the T lymphocyte data set.

(C) *Inclusion of all data points*. All data points were initially included in analysis, with the exception of genes not expressed in any of the control samples (cDNA, tRNA, 10-cell samples). This excluded 12 genes from analysis in the neutrophil data set and 13 in the lymphocyte data set. In addition, any transcripts missing from all samples in an array were excluded. This excluded 3 genes in the neutrophil set, for a total of 15 excluded

genes. No cells were excluded. LOD Ct was subtracted according to the $\text{Log}_2\text{EX}$ method as described above. Expression was evaluated for 81 genes in 220 cells for the neutrophil data set and 85 genes in 247 cells for the T lymphocyte data set.

## Data normalization

Three different sets of criteria were used for data normalization for each of the data sets generated from the three methods for data exclusion.

(1) *Mean centering.* The mean Ct value for each cell was calculated and subtracted from each data point for the same cell. This approach removes the dependence of magnitudes, allowing for easier visualization and comparison of relative differences in expression levels.

(2) *Quantile normalization.* Gene expression data for each cell was re-ordered by raw Ct value, and mean Ct values for each cell were calculated. The original data was then replaced by the average quantile, such that the highest value was replaced by the mean of the highest values, the second highest value by the mean of the second highest values, and so on. This method of rank-order analysis eliminates cell-to-cell differences in data density, by making the data distributions identical.

(3) *Standardization of the genes.* Gene expression data were mean-centered for each cell, and then the values for each gene were standardized (converted to *z*-scores) by mean-centering and dividing by the standard deviation. Residuals from an ANOVA with Plate as the main effect were extracted. This method adjusts the distribution only of targets whose expression differs among plates. A further centering of residual expression values to a mean of zero for each cell ensures that no cells have artificially low or high expression of all genes.

## Analysis of gene expression patterns

The single cell transcript abundance distribution for each gene was determined using SAS JMP Pro 10 (Cary, NC). For each gene, several models, including Normal, Gamma, Johnson Su, Johnson SI, Lognormal, and Weibull, were tested in order to find the model best fitting the data. Modality was assessed for the two best fitting models by Akaike information criteria (AIC) score and was further verified by calculating deltaAIC, comparing scores of bimodality, trimodality and unimodality, as well as visual observation. Genes exhibiting bimodality were tracked and cluster membership was determined in the raw data set as well as after data exclusion and normalization methods deemed most suitable, using the criteria above, namely exclusion by missing data cutoff and normalization by standardization of the genes. In addition, the number of cells included in each cluster was determined for each donor. Known gene product functionality was obtained from three data bases: ToppFun (*Chen et al., 2009*), DAVID v6.7 (*Huang da, Sherman & Lempicki, 2009a*; *Huang da, Sherman & Lempicki, 2009b*), and KEGG Pathway (*Nakao et al., 1999*).

## Analysis of donor-to-donor variability

For each donor, the cell count was determined for each of the cell clusters defined within the population across all donors. Following this, the observed frequencies were compared

to expected frequency by Chi-square test comparison of the number of cells of each class in each of the five individuals relative to the expectation assuming equivalent proportions.

## Comparison of primary analysis methods by concordance of cell clusters

Combining the methods for data exclusion and normalization generated nine alternate sets of processed data for each of the two cell types. Each data set was organized by hierarchical clustering as well as $k$-means clustering by cell, resulting in cell clusters based on shared gene expression patterns. Concordance, defined as the percentage of cells ascribed to the same cluster, was compared between all combinations of analysis methods for both methods of clustering. For hierarchical clustering, data was clustered using Ward's minimum variance method (*Ward, 1963*), which minimizes the total within-cluster variance using the total within-cluster sum of squares, under the assumption that distances between individual objects are proportional to Euclidean distance. The $k$-means method of clustering aims to sort data into a pre-defined number of clusters, $k$, with each data point belonging to the cluster with the nearest mean (*MacQueen, 1967*). $K$-means clustering was performed on all data sets with $k$ values of 2 or 3 for both neutrophils and T lymphocytes. The $k$ values were evaluated using Cubic Clustering Criteria (CCC) with external cluster validation. All computations were performed in SAS JMP-Genomics v5.0 (Cary, NC).

## RESULTS

### Gene expression pattern analysis

Gene expression analysis of the raw neutrophil data revealed the existence of different expression patterns for genes, such as unimodal distribution of expression (Fig. 2A), bimodal distribution with or without the existence of low expressors ($Ct_{35}$–$Ct_{39}$) (Figs. 2B and 2C), and trimodal distribution (Fig. 2D). The existence of non-expressing cells poses the problem of how to define these data points. One approach is to assign all such values the maximum Ct of 40, but this assumes that these data represent true missing expression; they could also result from technical errors due to failed PCR reactions. If the latter is the case, apparent bimodality with on/off expression patterns would in reality represent unimodal distribution with missing data points being technical artifacts instead of biologically relevant information. An alternative approach for addressing this issue is to look at patterns of missing data within the sets. If missing data points from the same genes tend to correlate within the cells, the cause is likely to be biological, suggesting that the populations contain cellular subgroups. In order to determine whether the existence of genes with bimodal expression patterns signaled the existence of cellular subclasses, the data was clustered based on shared gene expression patterns. Clustering showed that for neutrophils, bimodal genes exhibiting on/off pattern tended to be off in the same cells, although they clustered together with unimodal genes implying that the differential expression between cell types is not restricted to bimodality (Fig. 3A). Another potential cause for missing data points is low initial concentration of RNA in the sample, owing

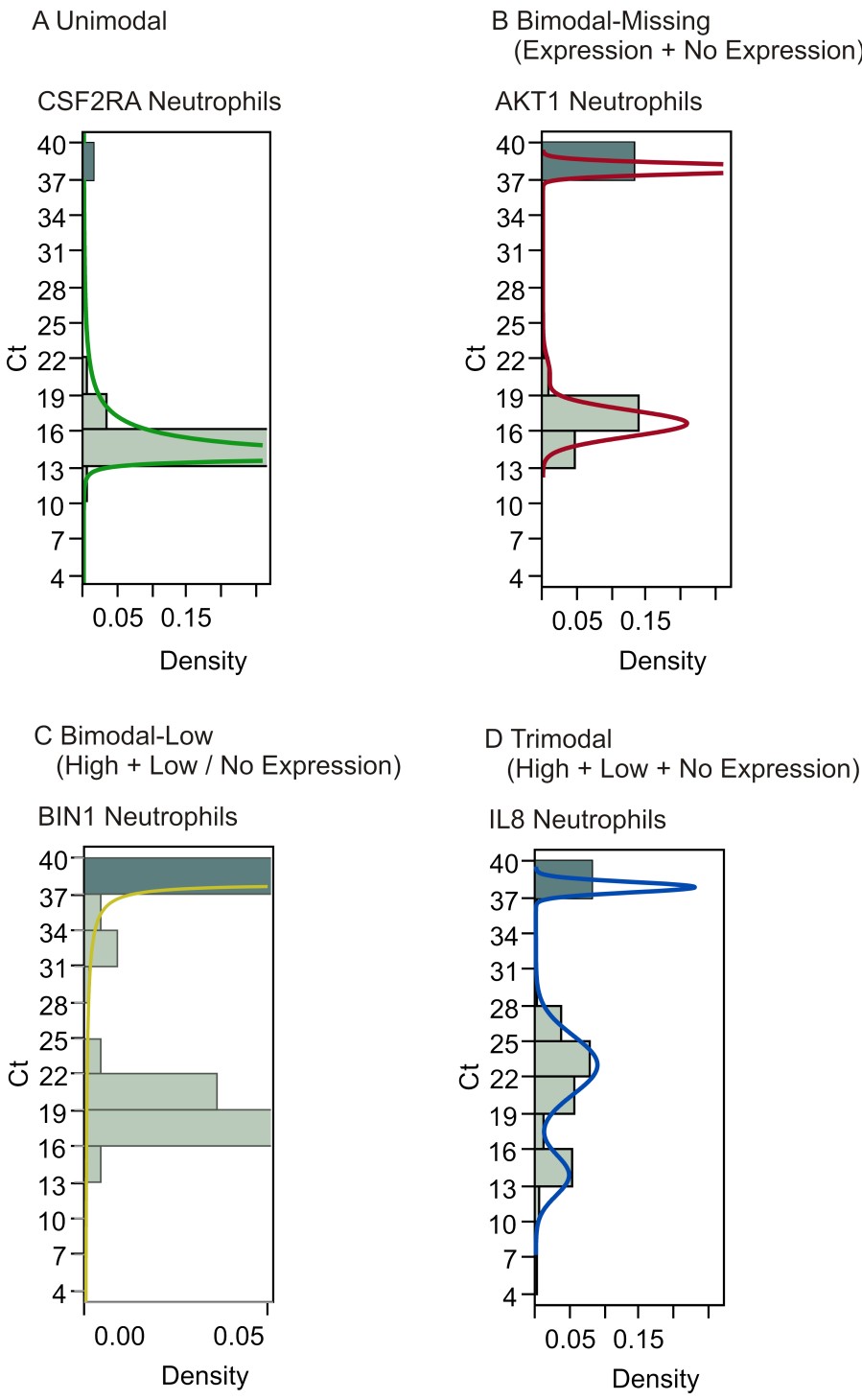

**Figure 2 Gene expression analysis show multimodal expression patterns.** Analysis of gene expression revealed varying patterns in both the neutrophil and T lymphocyte data sets. Examples from the neutrophil data set show (A) unimodal distribution, (B) bimodal distribution with one peak consisting of missing values i.e., Ct 40, (C) bimodal distribution with one peak consisting of both missing values and low expression, and (D) trimodal distribution. A peak at Ct 40 indicates the existence of cells showing no expression of the gene.

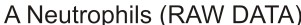
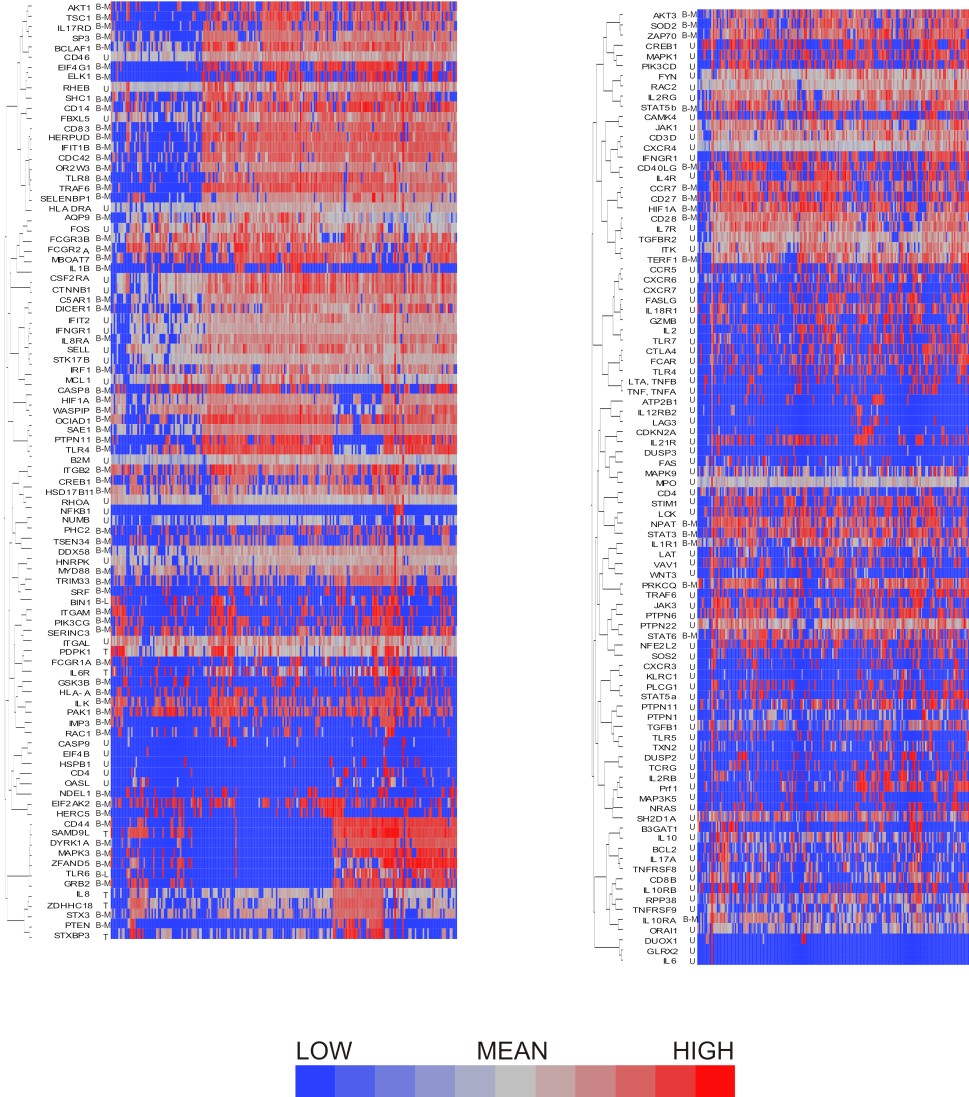

A Neutrophils (RAW DATA)     B T Lymphocytes (RAW DATA)

LOW     MEAN     HIGH

**Figure 3 Hierarchical clustering of neutrophil and T lymphocyte data showed distinct sub-populations of cells characterized by shared patterns of gene expression.** Hierarchical clustering of the pre-processed (A) neutrophil and (B) T lymphocyte data prior to data exclusion and normalization show bimodal genes preferentially clustering together. Bimodal genes are indicated by B-M (indicating that one peak consists only of cells with missing values for the gene) or B-L (indicating that one peak consists both of missing values and of cells with low expression). Unimodal genes are indicated by U, and trimodal genes by T. Dendrograms for columns not shown.

to inefficient RNA extraction, leading to complete loss of signal for the lowest-abundance genes that share the technical inefficiency. In order to address this, we controlled for overall abundance of RNA by normalizing our data sets.

Similarly to the neutrophils, the T lymphocyte data set contained genes exhibiting bimodal gene expression. As seen in neutrophils, T lymphocyte genes with bimodal on/off

expression patterns also tended to be interspersed with unimodally expressed transcripts (Fig. 3B).

## Detection of neutrophil subtypes

Hierarchical clustering was applied to the datasets using Ward's method, which has been shown to discriminate clusters efficiently on gene expression datasets (*Ferreira & Hitchcock, 2009*; *Ma & Zhang, 2012*). Figure 4 shows the results of hierarchical clustering with nine different methods combining the three methods for data exclusion and three methods for data normalization. The color coding (purple, green and orange) shows the degree of concordance of clustering relative to the method based on supervised data removal with mean centering (top left). Employing exclusion with any of the three methods, followed by either mean centering or quantile normalization, three clusters of neutrophils were observed consistently, with notable separation of the orange, and most of the green, clusters from the purple one. Concordance, defined as the percentage of cells assigned to the same cluster, ranged from 75% to 100%, *prima facie* supporting the presence of three cell types in our samples.

However, when a Fluidigm array plate effect was fit to the standardized gene expression $z$-scores, only two major clusters were observed regardless of the data exclusion method (Fig. 4G), and concordance of the two-way classification of orange versus green/purple cells was perfect. This analysis implies that a plate effect caused the splitting of the large purple/green clusters observed with the mean-centering and quantile normalization methods. That is to say, very low abundance gene expression led to loss of signal on one of the plates, generating an artificial signature of co-regulation of some cells. However, the orange cluster remains robustly detected by all methods. We conclude that there are two main clusters of cell types in neutrophils. There is also a hint of a sub-type within the orange cells defined by differential expression of a half-dozen genes, but a larger sample will be required to validate this inference.

## Hierarchical clustering verified the existence of cellular subgroups

Having compared methods for data exclusion and normalization, we opted to focus on the analysis method using a two standard deviation cutoff for exclusion with normalization by standardization of the genes (Fig. 4H). Hierarchical clustering revealed 2 major subclasses in both neutrophils (Fig. 5A) and T lymphocytes (Fig. 5B). The more clear definition of neutrophil subgroups, as compared to T lymphocytes, could be due to different levels of bimodality in the gene sets, such that more bimodality in the neutrophil data set gives rise to more distinct cellular subclasses. Alternatively, the two data sets could incorporate the same level of overall bimodality but differ in the level of co-variation of bimodally expressed genes. Since the expression of many genes on the T-cell array was too low to detect consistently, the analysis is based on fewer genes which also reduces the power to detect clusters.

More refined clustering of the T-cell data was also heavily impacted by the decision as to how to handle missing data. Including genes in the analysis according to the

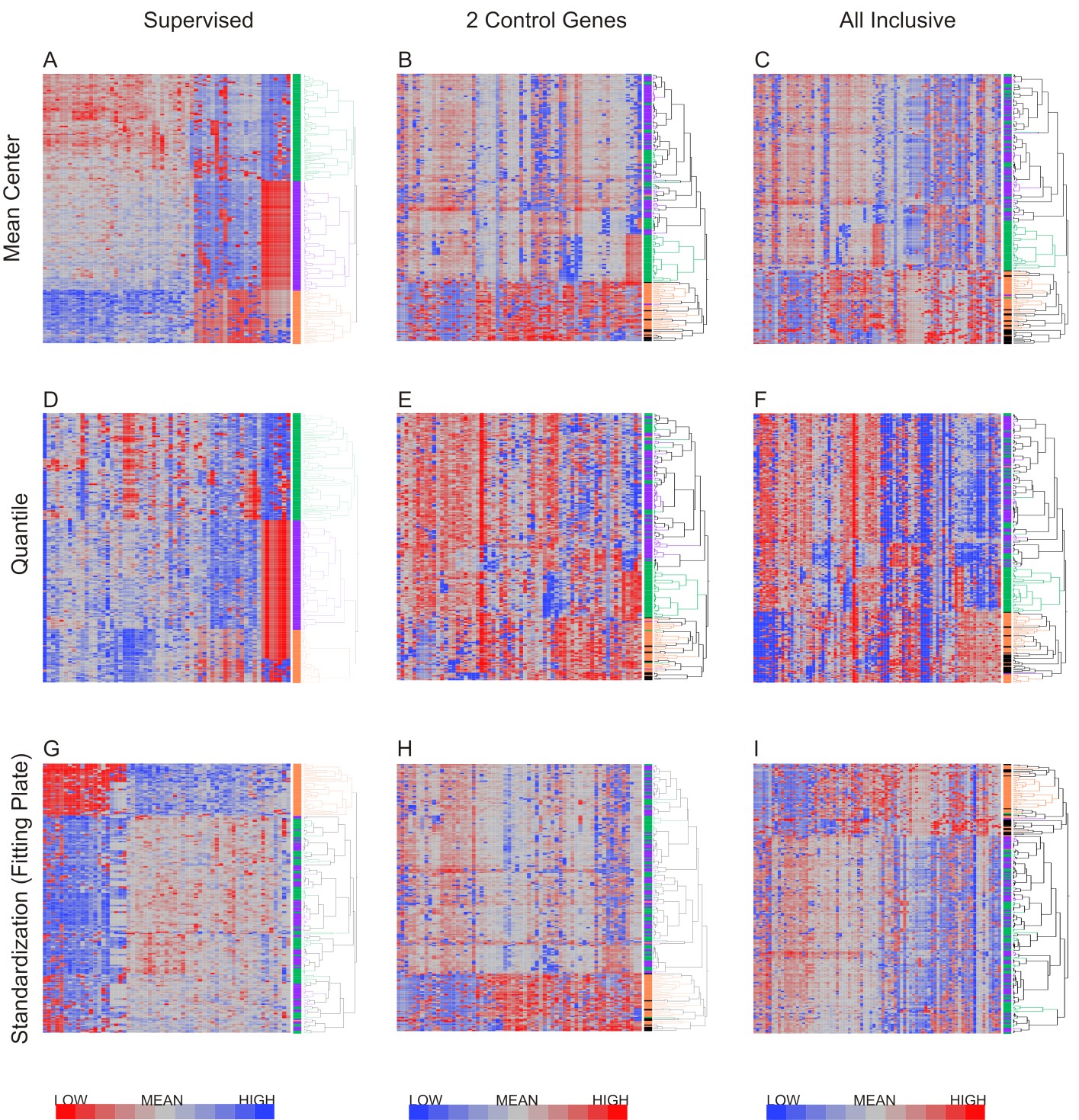

**Figure 4 Hierarchical clustering of neutrophil data after nine combinations of primary analysis.** Data was processed by 3 alternate methods of data exclusion (columns) and 3 methods of data normalization (rows). Following this, all resulting data sets were subjected to hierarchical clustering by Ward's minimum variance method. The results illustrate the effect of primary analysis method on data interpretation. Cells are colored by cluster for data analyzed by exclusion based on the supervised method and normalization by mean centering (top left heatmap). Dendrograms for columns are not shown.

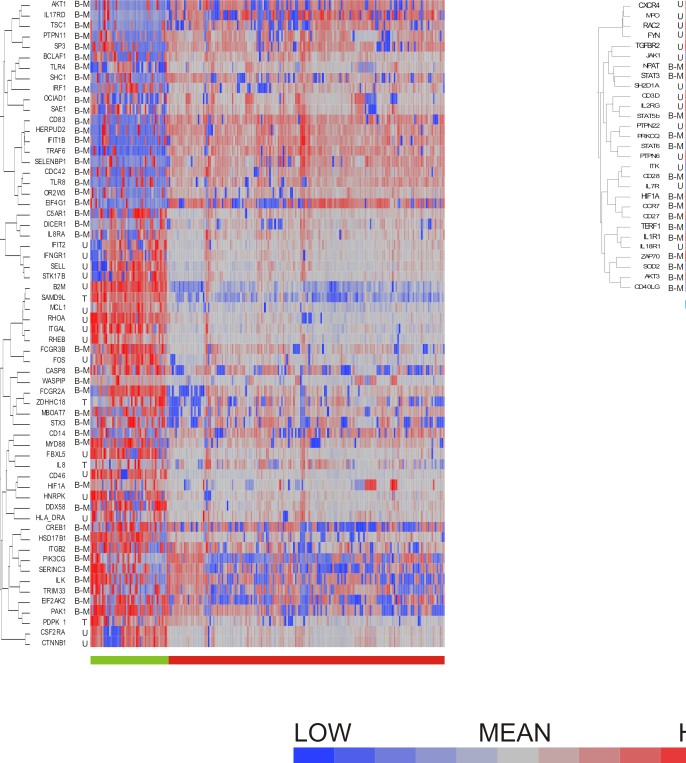

A Neutrophils    B T Lymphocytes

LOW        MEAN        HIGH

**Figure 5  Distribution of bimodal genes in hierarchical clusters after primary analysis.** Hierarchical clustering of processed (A) neutrophil and (B) T lymphocyte data after data exclusion by standard deviation cutoff and normalization by standardization of genes resulted in cell clusters defined by shared gene expression patterns. Genes that were multimodal before primary analysis are indicated. Bimodal genes are indicated by B-M (indicating that one peak consists only of cells with missing values for the gene) or B-L (indicating that one peak consists both of missing values and of cells with low expression). Unimodal genes are indicated by U, and trimodal genes by T. Dendrograms for columns are not shown.

2 standard deviation cutoff, setting missing data to null expression resulted in 6 clusters of cells irrespective of the data normalization procedure. In contrast, when missing data was assumed to be due to technical error and thus assigned the mean value for that transcript, the number of cellular subgroups observed after clustering differed: mean centering resulted in 2 large and 6 small clusters, quantile normalization in 7 clusters, and standardization of the genes in 6 clusters. The all inclusive method of data exclusion also resulted in differing numbers of cell clusters depending on the normalization method, with mean centering indicating 4 cellular subgroups, while quantile normalization showed five groups, and standardization of the gene resulted in six groups after hierarchical clustering by visual observation. Concordance of cell clustering across methods ranges between 70% and 80%, arguing that there are multiple cell states despite the high degree of heterogeneity. Concordance of the two-state clustering indicated in Fig. 5B was 95%.

**Table 1 Distribution of neutrophils between clusters shows donor to donor variability.** Distribution of cells is shown as observed number of cells per donor and cluster, as well as for the combined donor population. (A) Neutrophils after supervised data exclusion. (B) Neutrophils after data exclusion by median standard deviation cutoff. (C) T lymphocytes after data exclusion by median standard deviation cutoff.

**(A) Neutrophils with 2 clusters (Supervised STD)**

| Cell count | Donor 1 | Donor 2 | Donor 3 | Donor 4 | Donor 5 | Total cell count per cluster |
|---|---|---|---|---|---|---|
| Cluster A | 4 | 10 | 2 | 17 | 6 | 39 |
| Cluster B | 40 | 31 | 38 | 21 | 33 | 163 |
| Total cell count per donor | 44 | 41 | 40 | 38 | 39 | 202 |

**(B) Neutrophils with 2 clusters (2G STD)**

| Cell count | Donor 1 | Donor 2 | Donor 3 | Donor 4 | Donor 5 | Total cell count per cluster |
|---|---|---|---|---|---|---|
| Cluster A | 40 | 30 | 39 | 21 | 33 | 163 |
| Cluster B | 5 | 10 | 1 | 17 | 12 | 45 |
| Total cell count per donor | 45 | 40 | 40 | 38 | 45 | 208 |

**(C) T Lymphocytes with 2 clusters (2G missing STD)**

| Cell count | Donor 1 | Donor 2 | Donor 3 | Donor 4 | Donor 5 | Donor 6 | Total cell count per cluster |
|---|---|---|---|---|---|---|---|
| Cluster A | 17 | 33 | 44 | 34 | 28 | 42 | 198 |
| Cluster B | 9 | 10 | 1 | 2 | 20 | 4 | 46 |
| Total cell count per donor | 26 | 43 | 45 | 36 | 48 | 46 | 244 |

## Individual differences in donor representation in cellular subgroups

We next turned to analysis of differences in cellular abundance among donors, and asked whether cells from all donors were equally distributed among the observed clusters. The results show that the frequency of cells in each neutrophil cluster differed between donors (Tables 1A and 1B) with donor 3 having a significantly lower than expected proportion of cells in cluster A, whereas donor 4 has the inverse profile. Setting the number of clusters to 2 in the analysis of the standardized data following supervised normalization, the $\chi^2$ value for differences in cell type abundance is 24.5 ($p = 6 \times 10^{-5}$, 4 degrees of freedom) (Table 1A).

As the intrinsic variability of T cell populations is greater than that of neutrophil populations, it is perhaps not surprising that we found donor-to-donor variability to be larger for T cells than for neutrophils. Compared to neutrophils, T cells had considerable variability in cell distribution between subgroups. The counts associated with the smallest subgroup were not large enough to establish whether the donors differ, but they do suggest divergence for the other clusters. Setting the number of clusters to 2 following data normalization, the $\chi^2$ value for differences in cell type abundance is 36.8 ($p = 7 \times 10^{-7}$,

5 degrees of freedom) (Table 1C). Sampling of more cells in more donors will be required to establish whether these differences correlate with physiological and immunological attributes of the individuals.

## DISCUSSION

To determine whether variation in gene expression correlated with variation in cellular phenotype, gene expression data for all genes were analyzed across all cells for expression patterns, such as unimodality and bimodality. Patterns differing from the prevalent, long-tailed, log-normal distribution may reflect active processes that contribute to cell–cell variation and thus functional subclasses of cells (Dalerba et al., 2011). Bimodality, in particular, can be expected in immune cell populations, due to the possibility that cells within such a population may be in states of either pre- or post-activation, with the changes in gene expression that this would entail. While bimodal behavior is a potentially important feature of gene expression in a population and can reflect true differences between subpopulations (Shalek et al., 2013), not all bimodal distributions are likely to reflect biological reality in an unprocessed single-cell data set. The risk of excluding true bimodality by setting the cutoff too low must be weighed against the risk of including artificial bimodality by inclusion of all data points and thus more measurement-derived noise. In addition, it is desirable to differentiate between bimodality due to high versus low expression of a given gene and bimodality due to a gene being expressed or not expressed. Finally, technical artifacts such as plate effects can also induce apparent bimodality if expression of low-abundance transcripts drops out completely in one plate.

The occurrence of bimodality of gene expression in both neutrophils and T cells leads us to conclude that the cell populations tested contain specific cellular sub-types. The results show unambiguous evidence for two cellular subtypes in both the neutrophil and T-cell populations, possibly with additional subtypes that will require larger datasets to validate. The nature of the bimodal genes involved, however, hint at the functional nature of the cellular subgroups. For example, the neutrophil cluster represented by low TLR4/8, high PAK1, high ITGB2 (subunit of LFA-1) profiles would likely occur when extravasation and cell motility is more essential than direct microbial phagocytosis.

Methods for analyzing population level data are well established; however these are not optimal for single cell data due to the high variability of gene expression between individual cells and the intrinsic noise in single cell data sets. Gene expression levels, even of housekeeping genes, can differ 1000-fold between individual cells (Bengtsson et al., 2005), and analysis of individual single-cell PCR calibration curves do not produce reliable values (Liss et al., 2001). In order to overcome this issue, independent measurements by alternative methods such as RNA FISH (Bajikar et al., 2014; Shalek et al., 2013; Wang, Brugge & Janes, 2011), smFISH (Rahman & Zenklusen, 2013), or immunochemistry (Dalerba et al., 2011) can be used to verify select targets; however, these do not allow for easy verification across large target sets at the individual cell level. Comparison of the outputs from the different methods of primary analysis tested illustrates the impact of analysis method on subsequent interpretation of biologically relevant information such

as cellular subtypes within a population. Our recommendation is to use standardization methods that allow for fitting of technical effects, such as the plate effect that generated two sub-types in the mean-centering and quantile normalization strategies. Data exclusion should be aware of the possibility that missing data reflects technical failure, but for the most part it seems to be due to very low and possibly missing expression. Replacement of missing data with average expression did not unduly impact our clustering at the 2-cell type level, and does not appear to be justified.

Cluster analysis is a natural choice for interpretation of qRT-PCR data. We employed two hierarchical clustering methods in order to quantitatively assess the robustness of our primary data processing methods. The results obtained by both methods of clustering were then compared, and the concordance between clusters, as defined by shared cluster assignment for cells, showed that $k$-means and hierarchical clustering approaches influence the conclusions but to a lesser extent than the data normalization strategy. The two approaches disagreed as more sub-types were added to the analysis, but were in good agreement at $k = 2$ cell types for both neutrophils and T-cells.

An additional question we addressed was whether or not the type of cell would have an effect on the concordance, in other words, whether different cell types would require different methods of data exclusion and normalization for optimized analysis outcome. It should be noted that although the trends are similar in both cell types, neutrophils show an overall lower heterogeneity than T cells. The observed higher stability of concordance of neutrophil clusters when compared to T cell clusters is likely affected by these inherent properties of neutrophil and T cell populations. It is thus important to consider not only cluster robustness when choosing analysis methods, particularly when data represents a heterogeneous population, such as the T lymphocyte population investigated here.

## CONCLUSION

Our study shows that using single cell analysis we can potentially detect functional subclasses not previously appreciated within immune cell populations. Bimodal patterns of gene expression within the cell populations suggested cellular subclasses, and this was confirmed by hierarchical clustering of cells. Emerging techniques enabling the study of single cell transcription levels have made clear the need for insight into the appropriate methods of analyzing the data generated. Our systematic testing of different methods of single cell data analysis clearly illustrates the differences in subsequent interpretation of the processed data. Importantly, our results highlight the necessity of using a method that adjusts for any defined technical effects, and that failure to do this will affect the inference of biological properties.

**List of abbreviations**

| | |
|---|---|
| **AIC** | Akaike information criterion |
| **Ct** | Cycle threshold |
| **k** | cluster count |
| **LOD** | Limit of Detection |
| **qRT-PCR** | Quantitative Real-Time PCR |

## ACKNOWLEDGEMENTS

The authors wish to thank Dr Dalia Arafat for help and advice with qRT-PCR data acquisition, Marisa Casola for assistance in sample acquisition, and The Center for Health Discovery and Well-Being at Emory Midtown Hospital in Atlanta for donor samples.

### Funding

Funding for this work was provided to GG and MLK by the Petit Institute of Bioengineering and Bioscience at Georgia Institute of Technology and NIH award DP2OD006483 to MLK. The funders had no role in study design, data collection and analysis, decision to publish, or preparation of the manuscript.

### Grant Disclosures

The following grant information was disclosed by the authors:
Petit Institute of Bioengineering and Bioscience at Georgia Institute of Technology.
NIH: DP2OD006483.

### Competing Interests

The authors declare they have no competing interests.

### Author Contributions

- Linda E. Kippner and Jinhee Kim conceived and designed the experiments, performed the experiments, analyzed the data, wrote the paper, prepared figures and/or tables, reviewed drafts of the paper.
- Greg Gibson and Melissa L. Kemp conceived and designed the experiments, analyzed the data, contributed reagents/materials/analysis tools, wrote the paper, reviewed drafts of the paper.

### Human Ethics

The following information was supplied relating to ethical approvals (i.e., approving body and any reference numbers):

Georgia Institute of Technology Institutional Review Board: H09364.

### Supplemental Information

Supplemental information for this article can be found online at http://dx.doi.org/10.7717/peerj.452.

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
