# Peer review of "Single cell transcriptional analysis reveals novel innate immune cell types"

_PeerJ, doi:10.7717/peerj.452_

## Round 0.1 · original submission · Major Revisions

Reviewer 2 raises a number of concerns that should be addressed. Along with the small number of questions, also consider the submission of supplemental data as mentioned by Reviewer 1.

Reviewer 1 ·

Basic reporting

• The text is reasonably clear and well organized.
• The Introduction cites many of the important and relevant papers in the field. However, there have been some more-recent publications that have emphasized the technical irreproducibility of single-cell transcriptomic measurements: PMID 24141095, 23306461, 24056876.
• The submitted manuscript is appropriately structured for this journal
• The figures are appropriate to the manuscript and are of sufficient resolution. However, for Figures 3 and 5, the dendrograms should be put on the opposite side of the transcript labels so that it is easier for the reader to match the column labels to the data. Also, transposing the clustergrams should be considered so that the reader does not have to rotate the page (or their head) 90 degrees to read the labels.
• The submission is definitely a self-contained unit of publication.

Experimental design

• The manuscript describes primary research and clearly defines the research question.
• The research question itself is relevant and meaningful, particularly given the enormous enthusiasm growing for single-cell methods (PMID 24524124).
• The experiments appear to have been conducted rigorously given the measurement platform selected by the authors.
• The methods are mostly described well enough to be reproducible by another group. The one exception is how the donor-to-donor variability was assessed. For “the cell clusters defined within the overall population” (Lines 221-2), is that referring to the population of each patient, or the populations defined across all patients? The latter grouping makes a lot more sense and would speak to the reproducibility of the clusters across patients.
• The human subject research was performed according to proper IRB approvals.

Validity of the findings

• Although not required for publication, it should be worth mentioning in the manuscript that the strongest validation of single-cell qPCR data is to confirm heterogeneity with an indepdendent, such as RNA FISH (PMID: 21873240, 24449900), smFISH (PMID: 23685454), or immunochemistry (PMID: 22081019). Flow cytometry would be especially suitable for blood subtypes.
• There is no indication of where the data will be deposited. Because there is not a standard repository, the easiest thing would be to include it as a supplemental data file for the manuscript.
• The conclusions are appropriately stated and are not speculative.

Comments for the author

• The authors do a good job reminding that “on/off” gene expression could simply be caused by irreproducible technical variation at the single-cell level (PMID 24141095, 23306461, 24056876).
• One question that is worth considering is making the distinction between clustering variations caused by the preprocessing compared to the general instability of hierarchical clustering. For example, if one transcript or single-cell observation were removed from the dataset, how much would the clusters change around? This would provide strong evidence that it is indeed the data inclusion and normalization that is affecting the results.

Reviewer 2 ·

Basic reporting

The authors have generated ~500 wells worth of Fluidigm single cell gene expression data from 5 healthy donors and two cell types: neutrophils and T cells. They considered 3 methods of data quality control: using heuristic cutpoints to throw out low-expression genes and/or wells; throwing out genes whose mean level of expression differed too much from a set of two reference genes and low expressing cells; and no QC. They then considered 3 methods of normalization: mean centering of each gene (thus no normalization); quantile normalization of each cell; and regression for plate effects. The authors argue throwing out genes whose mean level of expression differed too much from the reference genes and regression for plate effects maximize clustering, so are the most desirable techniques.

Experimental design

There isn't any biological ground truth to evaluate the QC and normalization approaches, so we are left with inventing our own. This is of course often the case in exploratory experiments, but I don't find the adoption of the “three clusters of T cells” observed in the as the ground truth to be very convincing or useful. How do we know that the three clusters of cells correspond to biologically meaningful units, as opposed to being due to technical factors? If we can't distinguish between these two possibilities, then picking the method that maximizes the clustering would actually be the worst option.
What about comparing the single cell readings to the 10 cell and 100 cell? Or see how closely each donor clusters with him or herself, or divide each donor into two sets and calculate a correlation coefficient between set A and set B for each donor. A good QC and standardization method ought to maximize any and all of these.

Validity of the findings

What model is being fit to data to test for bi/tri-modality? Since it's not otherwise specified, I assume a Gaussian mixture model. Based on the histograms, it does not appear that the cluster with Ct =40 is Gaussian at all—instead it's a spike of cells with undetected levels of transcript. Since the Gaussian model is mis-specified (ie, blatantly wrong), use of the AIC is suspect. I don't have a good solution to this, other than it should be addressed in the discussion and the results involving categorization of genes into bimodal/unimodal need to be weakened, because the authors' method is unlikely to be reliable in detecting bi-modality or a lack thereof.

Comments for the author

QC and Data Exclusion (page 6)
For the supervised data exclusion why only throw out low-expression genes? Maybe cells with low expression are similarly suspect? Is there any motivation for the 70/70 cutoff?

Data Normalization (page 7)
The description of the quantile normalization is unclear to me. The authors might wish to introduce some notation to make it easier to follow (ie, expression of cell i of gene j is E[ij]).
For the “standardization of genes” which plate effect was removed, the lysis plate, or the Biomark Chip, or both? Is it possible to estimate a lysis plate effect, without completely removing donor variability? Lysis plate appears to be correlated with donor, but I cannot tell to what degree the two are confounded.
For all methods, please explain how many cells/genes are being thrown out, perhaps with a 3 x3 table summarizing the combination of methods.

All heatmap figures
Are rows being clustered, if so how? Please indicate in the legends if clustering is occurring and the dendograms are just being suppressed to reduce clutter. Also, please indicate what metric (eg Euclidian distance?) was used to generate the distance matrix between rows/columns.

Figure 3: It seems like almost all genes are at least somewhat “bimodal”: there are isolated patches of blue (no expression present). And just because columns are proximal to each other, doesn't necessarily mean that genes are close to each other in distance, eg, NDEL1/EIF2AK2 in fig 3A.

If missing data points from the same genes tend to correlate within the cells, the cause is likely to be biological, suggesting that the populations contain cellular subgroups. (Page 9, line 253)

I don't think this conclusion necessarily follows, as the authors note two sentences below:
Another potential cause for missing data points is low initial concentration of RNA in the sample, owing to inefficient RNA extraction, leading to complete loss of signal for the lowest-abundance genes that share the technical inefficiency.

Ward's method references, page 9: are these references for single cell expression?

---

## Round 0.2 · accepted · Accept

Thank you for fully addressing the reviewers' questions and comments.

Reviewer 2 ·

Basic reporting

I am happy with the revision and have no further comments.

Experimental design

I am happy with the revision and have no further comments.

Validity of the findings

I am happy with the revision and have no further comments.

Comments for the author

I am happy with the revision and have no further comments.